# FLEX: Unifying Evaluation for Few-Shot NLP

Jonathan Bragg[*]    Arman Cohan[*]    Kyle Lo    Iz Beltagy

Allen Institute for AI, Seattle, WA

{jbragg,armanc,kylel,beltagy}@allenai.org

## Abstract

Few-shot NLP research is highly active, yet conducted in disjoint research threads with evaluation suites that lack challenging-yet-realistic testing setups and fail to employ careful experimental design. Consequently, the community does not know which techniques perform best or even if they outperform simple baselines. In response, we formulate the FLEX Principles, a set of requirements and best practices for unified, rigorous, valid, and cost-sensitive few-shot NLP evaluation. These principles include Sample Size Design, a novel approach to benchmark design that optimizes statistical accuracy and precision while keeping evaluation costs manageable. Following the principles, we release the FLEX benchmark,[2] which includes four few-shot transfer settings, zero-shot evaluation, and a public leaderboard that covers diverse NLP tasks. In addition, we present UniFew,[3] a prompt-based model for few-shot learning that unifies pretraining and finetuning prompt formats, eschewing complex machinery of recent prompt-based approaches in adapting downstream task formats to language model pretraining objectives. We demonstrate that despite simplicity, UniFew achieves results competitive with both popular meta-learning and prompt-based approaches.

## 1 Introduction

Few-shot learning, the challenge of learning from a small number of examples, is critical for developing efficient, robust NLP techniques [71, 76]. In recent years, separate threads of few-shot NLP research have pursued goals like generalization to new classes [e.g., 5, 25], adaptation to new domains and tasks [e.g., 3, 4, 21], and direct application of pretrained language models (LMs) [e.g., 10, 24, 55, 56]. Unfortunately, despite the shared goal of advancing few-shot NLP techniques, the community does not know which techniques work best or even if they perform better than simple baselines. Evaluation suites across these research threads are disjoint, lack challenging-yet-realistic testing setups (e.g., class imbalance, variable training set sizes, etc.), and do not employ careful experimental design to ensure accurate and precise evaluation estimates and minimal computational burden. Prior work in few-shot learning outside of NLP serves as a stark warning of the consequences of improper measurement: Dhillon et al. [19] showed that techniques from several years of prior work did not make clear progress due to large overlapping accuracy distributions and, moreover, do not outperform a simple, carefully-tuned baseline.

**Need for systematic benchmark design** As such, a high-quality benchmark is urgently needed to enable rigorous comparison of techniques across disjoint, highly-active threads of few-shot NLP research. But what should such an evaluation suite look like? Some best practices for evaluation of few-shot methods have been introduced in the computer vision (CV) literature [19, 67] and should

---

[*]Equal contribution

[2]Benchmark, leaderboard, and benchmark creation toolkit: https://github.com/allenai/flex. Apache License 2.0

[3]Few-shot model: https://github.com/allenai/unifew. Apache License 2.0

Table 1: Comparison of the FLEX benchmark with closest prior work. Our benchmark consists of episodes with variable number of shots in the range [1-5] and with class imbalance. "No extra test data" refers to excluding validation data from testing tasks, to avoid unfairly advantaging models that use such data [50]. Our benchmark's number of test episodes is selected to balance statistical accuracy and precision, which suffers in few-episode setups, and compute requirements, which is too costly in many-episode setups (§5).

| | CrossFit[75] | LM-BFF[24] | GPT-3[10] | DS[5] | SMLMT[4] | FewGlue[56] | **FLEX (ours)** |
|---|---|---|---|---|---|---|---|
| Class Transfer | - | - | - | ✓ | - | - | ✓ |
| Domain Transfer | - | - | - | - | ✓ | - | ✓ |
| Task Transfer | ✓ | - | - | - | ✓ | - | ✓ |
| Pretraining Transfer | - | ✓ | ✓ | - | - | ✓ | ✓ |
| Shots per class | {16, 32} | 16 | variable | {1,5} | {4,8,16,32} | {total 32}[4] | [1–5] |
| Variable shots | - | - | ✓ | - | - | - | ✓ |
| Unbalanced | - | - | - | - | - | - | ✓ |
| Textual labels | ✓ | ✓ | ✓ | - | - | ✓ | ✓ |
| Zero-shot | - | ✓ | ✓ | - | - | - | ✓ |
| No extra test data | - | - | - | ✓ | ✓ | mixed[5] | ✓ |
| # test episodes | 5 | 5 | 1 | 1000 | 10 | 1 | 90 |
| Reporting | avg | avg, SD | avg | avg, SD | avg, SD | avg, SD | all[6] |
| # datasets | 160 | 16 | 37 | 7 | 18 | 8 | 20 |

be applied to NLP. However, unifying few-shot NLP work introduces new challenges. For example, the benchmark needs to test all types of transfer studied in separate research threads to measure progress on new techniques that make gains in each of these important generalization settings (§2). Also, given the importance of zero-shot learning and learning from task descriptions [29, 73], the benchmark needs to include zero-shot episodes and textual labels to enable measuring progress for models that do not use conventional supervised training, including methods that leverage the latent knowledge in pretrained LMs [10, 24, 78]. Further, the benchmark must accommodate new, computationally-expensive approaches, without overly reducing the number of evaluation episodes at the expense of statistical accuracy [3, 24, 75].

**Need for a robust few-shot model** Recent prompt-based models [10] have shown strong results in few-shot learning. These models leverage the power of (often large) pretrained language models and adapt the format of downstream tasks to the underlying pretraining objective (e.g., Masked Language Modeling). This way, given the right natural language prompt (and sometimes verbalizers [55] and additional demonstrative examples), the model can quickly fine-tune on the downstream task [24, 43, 44, 55, 56]. However, adapting task formats to the underlying (masked) language modeling objectives is not straightforward; such models have been shown to be sensitive to varying choices of the prompt/demonstrations, training settings, hyperparameters, and learning algorithms [33, 50, 78], often requiring large held out sets and/or complex methods to overcomes such challenges. Can models eschew complex prompt engineering by unifying pretraining and downstream task formats?

In this paper, we tackle these key issues by introducing FLEX—**F**ew-shot **L**anguage **E**valuation across (**X**) many transfer types—and contributing the following:

- FLEX Principles (§3), a set of requirements and best practices for few-shot NLP evaluation that enables unified, rigorous, valid, and cost-sensitive measurements.
    - Sample Size Design: In support of valid, cost-sensitive measurement, we introduce a novel approach to few-shot sample size design (§5) that optimizes for a benchmark's statistical accuracy and precision while keeping computational costs accessible to a broad range of researchers.
- FLEX benchmark (§4), an implementation of the FLEX Principles. It tests across *four* few-shot transfer settings,[7] and includes a public leaderboard for few-shot NLP that covers 20 datasets across diverse NLP tasks (e.g., NLI, relation classification, entity typing). Table 1 summarizes key differences between FLEX and other few-shot NLP evaluation suites.

---

[4] The total number of training shots in each episode, not number of shots per class per episode.
[5] Most users use unlabeled examples, though recently, Tam et al. [65] do not.
[6] Average (avg), confidence interval (CI), standard deviation (SD), individual episode metrics
[7] Prior work evaluated at most two settings.

- UniFew (§6), a prompt-based model for few-shot learning in NLP. While most existing methods leverage pre-trained LMs for few-shot learning, LM pre-training tasks do not closely match natural downstream task formats, requiring complex methods (e.g., extensive prompt-engineering, use of verbalizers, episodic hyperparameter tuning, custom learning algorithms) to make these models work in few-shot setting. Instead, the key idea of our model, UniFew, is to close the gap between pre-training and fine-tuning formats by posing tasks as multiple-choice QA and using an underlying model that is pre-trained on a similar natural QA task format. This eliminates the need for complexities of adapting downstream tasks to the LM objectives, while resulting in competitive performance with both recent few-shot and meta-learning methods.

To aid similar efforts, our release of FLEX includes a toolkit for benchmark creation and few-shot NLP model development, which we used to create the FLEX benchmark and train UniFew.

## 2 Background and Related Work

We first provide background and notation for few-shot learning and evaluation, then discuss related work in NLP and outside NLP that motivated us to create the FLEX Principles and benchmark.

**Few-shot background and notation**  Broadly, modern approaches to few-shot learning are evaluated in a three-phase procedure [68]. In the first phase, a general-purpose pretrained model is obtained. In the subsequent "meta-training" phase,[8] techniques aim to adapt the model to be well-suited for few-shot generalization. Finally, a "meta-testing" phase evaluates the adapted model in new few-shot prediction settings.

Let $\mathcal{D}$ be a dataset of $(x, y)$ examples with full label set $\mathcal{Y}_{\mathcal{D}}$. From it, we construct three *sets* of episodes, corresponding to meta-training, meta-validation, and meta-testing and denoted by $\mathcal{E}_{\text{train}}$, $\mathcal{E}_{\text{val}}$, and $\mathcal{E}_{\text{test}}$, respectively. Each episode in each of these sets is a few-shot problem with its own test set and other attributes. Formally, each episode $E$ is a tuple $(\mathcal{D}^E_{\text{train}}, \mathcal{D}^E_{\text{val}}, \mathcal{D}^E_{\text{test}}, \mathcal{Y}^E_{\mathcal{D}})$, where $\mathcal{Y}^E_{\mathcal{D}}$ is a sampled subset of labels in $\mathcal{Y}_{\mathcal{D}}$ and $\mathcal{D}^E_{\text{train|val|test}}$ are disjoint sets of examples from $\mathcal{D}$ with labels in $\mathcal{Y}^E_{\mathcal{D}}$.[9] For each episode, the model's objective is to correctly predict labels for examples $\mathcal{D}^E_{\text{test}}$. To accomplish this, models make use of labeled examples in $\mathcal{D}^E_{\text{train}}$, which is typically configured such that each label $i$ in $\mathcal{Y}^E_{\mathcal{D}}$ has $K^E_i$ provided examples; $K^E_i$ is known as the *shot*, and the setting when a class has no examples in $\mathcal{D}^E_{\text{train}}$ (i.e., $K^E_i = 0$) is called *zero-shot*.

**Few-shot evaluation in NLP**  Research in few-shot NLP has proceeded in several parallel threads, each focused on a different type of transfer ability [76]. Each thread has separate evaluation practices, and the vast majority of few-shot NLP research has limited evaluation to a single transfer type (see Table 1). Here, we describe these types of transfer and their evaluation practices.

Following the CV literature [67, 68], one thread of few-shot NLP focuses on **class transfer**, the problem of generalizing from a supervised set of classes at meta-train time to a different set of classes from the same dataset at meta-test time. Evaluation typically involves splitting classes $\mathcal{Y}_{\mathcal{D}}$ into $\mathcal{Y}^{\mathcal{D}}_{\text{train}}$, $\mathcal{Y}^{\mathcal{D}}_{\text{val}}$ and $\mathcal{Y}^{\mathcal{D}}_{\text{test}}$ disjoint subsets. Class transfer has been studied on many text classification tasks [5], including relation classification [25, 28, 64], intent classification [37, 64], inter alia. In contrast, **domain transfer** keeps the same classes between meta-training and meta-testing but changes the textual domain (e.g., generalizing from MNLI to science-focused SciTail [4, 21]). Evaluation then requires identifying pairs of datasets with the same classes $\mathcal{Y}_{\mathcal{D}}$, where one dataset's episodes are assigned to $\mathcal{E}_{\text{train}}$ and the other's to $\mathcal{E}_{\text{test}}$. Domain transfer has also been studied on many tasks [3, 4], including dialogue intent detection & slot tagging [31], sentiment classification [77], NLI [21], and machine translation [27, 58].

Researchers have also begun to study **task transfer**, the problem of generalizing from a set of tasks at meta-train time to unseen tasks at meta-test time. Evaluation requires tasks (e.g., NLI) appearing in $\mathcal{E}_{\text{test}}$ *not* to appear in $\mathcal{E}_{\text{train}}$ or $\mathcal{E}_{\text{val}}$. Prior work has used GLUE tasks [70] for meta-training before meta-testing on tasks such as entity typing [3, 4], while other work instead used GLUE for

---

[8]Meta-training may include a "meta-validation" component, for validating generalization.

[9]In the few-shot literature, $\mathcal{D}^E_{\text{train}}$ and $\mathcal{D}^E_{\text{test}}$ are also called the *support* and *query* sets, and $|\mathcal{Y}^E_{\mathcal{D}}|$ the *way*.

meta-testing [21]. Very recent work has studied task transfer over a large set of datasets [75, 80]. A limited amount of work evaluates both domain and task transfer [3, 4, 21]. An important emerging line of work (not noted by Yin [76]) is **pretraining transfer**, the problem of whether pretrained language models can perform well at meta-test time without any meta-training. Evaluation in this setting requires $\mathcal{E}_{\text{train}}, \mathcal{E}_{\text{val}} = \emptyset$. Prior work has shown that pretrained language models are capable of surprising performance on many few-shot tasks, even without fine-tuning [10]. More recent work, mainly focusing on text classification, has reported further gains with cloze-style formats [55, 56, 65], prompt engineering [24], or calibration [78]. FLEX is designed to exercise all four of these transfer types from previous work.

**Few-shot evaluation outside NLP** The few-shot learning literature has largely focused on image classification, with the introduction of increasingly complex meta-learning algorithms [e.g., 23, 39, 54, 61, 68]. However, more recent work has shown that simple fine-tuning baselines are in fact competitive, and attribute this delayed discovery to problematic evaluation methodology [15, 19]. FLEX adopts recommended methodology [19, 67], and we introduce an analogous baseline (UniFew) to provide a strong measurement foundation for few-shot NLP.

# 3 FLEX Principles for Few-Shot NLP Evaluation

We now enumerate key desiderata for a few-shot NLP benchmark capable of solving the urgent problems with few-shot NLP evaluation, including separate evaluations for each transfer type and failure to incorporate best measurement practices from other domains (§2).

**Diversity of transfer types** To make NLP models broadly useful, few-shot NLP techniques must be capable of class, domain, and task transfer. Moreover, techniques should make use of the relevant supervision provided during meta-training to increase performance compared to the pretraining transfer setting. The benchmark should measure all four transfer settings to ensure that the community develops techniques that improve on strong pretraining transfer baselines, and enable comparison across these currently separate threads of research.

**Variable number of shots and classes** To better simulate a variety of real-world scenarios, the benchmark should include a variety of training set sizes and numbers of classes [67]. Testing robustness to these factors is crucial; few-shot techniques are often sensitive to changes in these factors [12], yet all prior few-shot NLP evaluations we are aware of used a fixed number of training shots and classes, known in advance during meta-training.

**Unbalanced training sets** The benchmark should also include unbalanced training sets with different training shots per class, another realistic setting adopted by CV benchmarks [67]. Class imbalance has also been observed to degrade performance [11, 47], yet prior few-shot NLP evaluations do not include this setting either.

**Textual labels** While numerical label values are often used in classification tasks, descriptive textual labels are also present for many tasks. Making these textual labels available for use by few-shot techniques enables the development of techniques that can leverage the class name, like in-context learning [10], template generation [24], and meta-learning [45]. Textual labels are crucial in particular for zero-shot evaluation.

**Zero-shot evaluation** We believe zero-shot evaluation is integral to the goals of few-shot evaluation. Similar to the motivation for measuring pretraining transfer, zero-shot evaluation is an important use case and also provides a strong baseline for some tasks. In the absence of training examples, textual class labels or richer task descriptions [73] must be provided. Some recent few-shot NLP work [e.g., 10, 24] evaluated with zero training shots, but most [e.g., 3, 5, 75] did not.

**No extra meta-testing data** We believe the benchmark should *not* provide validation data ($\mathcal{D}_{\text{val}}^E = \emptyset, \forall E \in \mathcal{E}_{\text{test}}$) or unlabeled data for meta-testing tasks, since few-shot learning seeks to enable high performance in environments where collecting additional data is costly.[10] Variation in these dimensions in prior NLP work makes comparison of results extremely difficult because it is often under-reported and gives unfair advantage to approaches that leverage such data [50]. For example, per-episode hyperparameter tuning on extra data has been shown to greatly inflate evaluation scores [24]. A few researchers [5, 65] follow our suggested approach, but others have used many

---

[10]Unlabeled data collection can be costly too, e.g. due to manual filtering [16].

different settings, from validation sets of various sizes [10, 24, 79] to no validation set but a large set of unlabeled examples [55, 56].

**Principled sample size design** Promising few-shot techniques can incur significant computational cost per episode, e.g., due to fine-tuning model parameters [4], searching for prompts [24], inter alia. To alleviate these costs, related works often evaluate with a limited number of episodes, which precludes statistically accurate or precise performance estimates. We believe the benchmark's test sample size should be optimized to enable proper performance evaluation for such techniques, while ensuring the computational burden is inclusive toward researchers without large compute resources.

**Proper reporting of CIs, SDs, and individual results** The benchmark should report confidence intervals (CIs) of performance estimates and follow recent guidelines [19] to report standard deviations (SDs) for understanding variability. Moreover, we newly advocate for controlling for the *same* sampled few-shot episodes across all methods and reporting individual episode results, so that researchers can run higher-powered paired statistical tests when comparing results [22], crucial when the benchmark has been optimized for low evaluation budgets.

## 4 FLEX Benchmark

The FLEX benchmark is a unifying, rigorous evaluation suite for few-shot learning in NLP, which implements the desiderata outlined in the previous section. In this section, we describe detailed design decisions and our accompanying few-shot NLP toolkit (§4.4), which we are releasing to facilitate easily adding NLP datasets and advanced sampling options to future benchmarks. We also describe the FLEX leaderboard (§4.5).

### 4.1 Task and Dataset Selection

Following GLUE [70] and other prior work [3, 5, 24, 78], we focus on tasks formatted as classification. Despite recent advances, NLP state-of-the-art models remain significantly worse than human performance on many text classification tasks, particularly in the few-shot setting. Automatic scoring of classification tasks is also more reliable than text generation tasks.

We selected datasets across three recent few-shot NLP evaluation suites, which separately studied class transfer [5], domain and task transfer [3, 4], and pretraining transfer [24]. Our benchmark includes a broad mix of tasks (NLI, question classification, entity typing, relation classification, and sentiment analysis) and formats (document, sentence, sentence pair). More complete dataset and license details are available in the following subsection and Appendix A.

### 4.2 Meta-Evaluation Protocols

As discussed earlier, FLEX evaluates four different types of transfer: Class, Domain, Task, and Pretraining Transfer. To support all types, we report results to the FLEX benchmark both *without* meta-training (pretraining-only) and *with* meta-training. This reporting scheme evaluates the performance of the basic pretrained model and the benefit (or lack thereof) of meta-training. A similar reporting scheme was proposed by Triantafillou et al. [67] for CV.

**Pretraining-Only** In this setting, the pretrained model is directly meta-tested on our benchmark without any additional training. This is the *Pretraining Transfer* setting, and it is the most difficult, but given the recent success of pretrained models in NLP for few-shot learning [10, 24], we believe that comparison to models without any meta-training is important for NLP tasks.

**Meta-Trained** In this setting, the model is meta-trained then meta-tested on our benchmark. We carefully selected and split datasets across meta-train/validation/test in order to enable testing of Class, Domain, and Task transfer with a single meta-training phase (to reduce computational burden). Datasets involved in each transfer setting (detailed split information in Table 4 in Appendix A):

- *Class Transfer*: FewRel [28], HuffPost [46], Amazon [30], 20News [38], and Reuters [41] take part in meta-training and meta-testing but with different classes.
- *Domain Transfer*: MR [49], CR [32], SNLI [9], and SciTail [35] are only in the meta-testing phase, but the corresponding sentiment and NLI datasets exist in the meta-training phase (MNLI [74], QNLI [52], and SST-2 [62]).

- *Task Transfer*: Subj [48], TREC [69], and CoNLL [66] are also for meta-testing only, and they represent tasks that the model does not encounter during meta-training.

Instead of per-episode hyperparameter tuning, we provide meta-validation episodes $\mathcal{E}_{\text{val}}$ for learning (during meta-training) global hyperparameters that work across all episodes. Specifically, the meta-validation dataset splits (see Table 4) consist of CoLa [72] for task transfer, WNLI [40] for domain transfer, and the validation splits used by Bao et al. [5] for all class transfer datasets. Following [3], we also include meta-training datasets MRPC [20], RTE [6, 8, 17, 26], and QQP [70].

### 4.3 Episode Sampling

We describe how our benchmark samples meta-testing episodes $\mathcal{E}_{\text{test}}$. For meta-training, we allow users to sample from $\mathcal{E}_{\text{train}}, \mathcal{E}_{\text{val}}$ in any way, or directly use the underlying dataset splits.

**Number of classes** For Class Transfer datasets, FLEX evaluates model robustness to variable number of new classes. When constructing episode $E$ from one of these datasets $\mathcal{D}$, our benchmark samples an episode-specific number of classes from dataset $D$, the sampler picks a random number from the range $\mathcal{Y}_{\mathcal{D}}^{E} \sim \text{Unif}(5, \min(|\mathcal{Y}_{\mathcal{D}}|, 10))$.[11] For Domain and Task Transfer, the number of classes is fixed to the maximum number of classes in each dataset because Class Transfer is not being evaluated.

**Number of shots** Following prior work outside NLP [47, 67], our benchmark samples the training shot independently for each episode $E$ and class $i$, as $K_i^E \sim \text{Unif}(K_{\min}, K_{\max})$, where $K_{\min} = 1$. Given strong performance of NLP models with few or even zero examples [10, 73] and following prior work [5], we set the limit $K_{\max} = 5$. Separately, we allocate an equal number of episodes as zero-shot, where we instead set $\mathcal{D}_{\text{train}}^{E} = \emptyset$ (equivalently, $K_i^E = 0, \forall i$). In each episode, examples are sampled uniformly at random without replacement (but can be reused across episodes).[12] Following Triantafillou et al. [67], we select a testing shot that is balanced across classes and leaves roughly half of examples for sampling the training examples. The total number of episodes for each reported configuration (pair of dataset and either zero- or few-shot) is set to 90 using Sample Size Design (§5).

### 4.4 Extensible Toolkit for Benchmark Creation and Model Training & Evaluation

Alongside the FLEX benchmark, we release an extensible, highly-configurable Python toolkit, which we used to generate the benchmark, and train and evaluate our models. Unlike existing meta-learning frameworks (e.g., Torchmeta [18], learn2learn [2]), our framework makes available a wide range of community-contributed NLP datasets and utilities via HuggingFace Datasets [42].[13] Our code also provides advanced sampling utilities (e.g., for class imbalance), ensures reproducibility by checksumming generated episodes, and reports all recommended statistics.

### 4.5 Public Leaderboard

We provide public leaderboards for each of the meta-evaluation protocols: Pretraining-Only[14] and Meta-Trained.[15] Submissions take the form of a text label predictions file, which is produced by our toolkit. Results are reported with confidence intervals, standard deviations, and individual predictions on request. See Appendix G for a screenshot of the results interface.

## 5 Sample Size Design: Balancing Statistical Measurement & Compute Cost

We demonstrate a principled approach to determining the optimal sample size configuration in our few-shot benchmark. A proper benchmark should produce performance estimates that are *accurate*, close to the true value, and *precise*, low variance. A large (test) sample size can achieve this, yet must be considered alongside computational cost so that a broad community of researchers with differing amounts of compute resources can participate. This decision is further complicated in the few-shot

---

[11]We limit to 10 classes to avoid undue burden on in-context approaches that fit examples in memory [10], and use a lower bound of 5 classes to match prior work [5].

[12]These samples represent an unbiased performance estimate, but do not eliminate underlying dataset biases.

[13]Apache License 2.0. Full license details for all software dependencies available in Appendix F.

[14]https://leaderboard.allenai.org/flex/

[15]https://leaderboard.allenai.org/flex_meta/

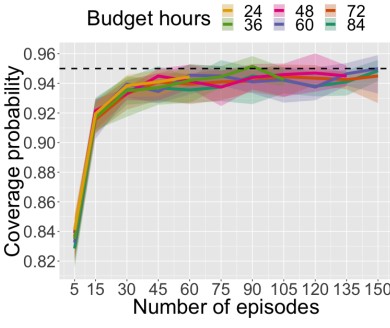
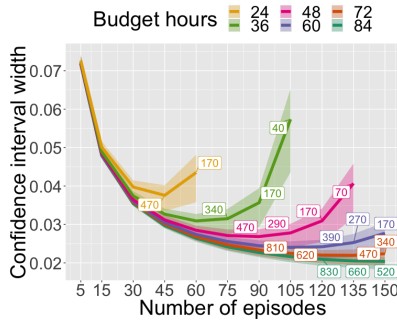

(a) Coverage probability of 95% CIs.          (b) Mean width of 95% CIs.

Figure 1: Results of simulation study described in §5. Each curve corresponds to a compute budget constraint $C$ (GPU-hours). Each point on a curve is an allocation of test data between the number of test episodes $|\mathcal{E}_{\text{test}}|$ or mean number of examples per episode $\overline{|\mathcal{D}_{\text{test}}|}$ such that evaluation can be completed within given budget. Per curve, lower values of $|\mathcal{E}_{\text{test}}|$ correspond linearly to larger values of $\overline{|\mathcal{D}_{\text{test}}|}$, which are shown as numerical text annotations in (b). Error bars represent the $10^{th}$ and $90^{th}$ percentile values from repeated simulations across $\mu_{acc} \in \{0.3, 0.35, \ldots, 0.95\}$.

setting, where sample size refers to both the number of test episodes $|\mathcal{E}_{\text{test}}|$ and the number of test examples $|\mathcal{D}_{\text{test}}^{E}|$ per episode $E \in \mathcal{E}_{\text{test}}$. For practicality, we consider $\overline{|\mathcal{D}_{\text{test}}|}$, the mean $|\mathcal{D}_{\text{test}}^{E}|$ across all episodes, rather than every $|\mathcal{D}_{\text{test}}^{E}|$. It remains unknown how one should best distribute test examples between $|\mathcal{E}_{\text{test}}|$ and $\overline{|\mathcal{D}_{\text{test}}|}$: More episodes each with fewer examples, or fewer episodes each with many examples? Prior work has been inconsistent in this regard. For example, Gao et al. [24] used $|\mathcal{E}_{\text{test}}| = 5$ and large $\overline{|\mathcal{D}_{\text{test}}|}$, while Bao et al. [5] used $|\mathcal{E}_{\text{test}}| = 1000$ and much smaller $\overline{|\mathcal{D}_{\text{test}}|}$.

Inspired by simulation techniques for informing statistically-powered experimental design [13], we study how different configurations of $|\mathcal{E}_{\text{test}}|$ and $\overline{|\mathcal{D}_{\text{test}}|}$ across different compute budgets $C$ impact the accuracy and precision of our estimated CIs, specifically with respect to *coverage probability* [53] and *width*. First, we estimate per-episode and per-test-example costs of our few-shot model (§6) to obtain valid $(C, |\mathcal{E}_{\text{test}}|, \overline{|\mathcal{D}_{\text{test}}|})$ configurations s.t. the full benchmark completes within given $C$ (GPU-hours).[16] Then, for each $(C, |\mathcal{E}_{\text{test}}|, \overline{|\mathcal{D}_{\text{test}}|})$, we perform 1000 simulation runs, in which each run samples predictions under a true model accuracy $\mu_{acc}$ and computes a single 95% CI, its width, and whether it correctly covers $\mu_{acc}$. Averaging over simulation runs gives us estimates for the coverage probability and width of our benchmark's CI for a single $(C, |\mathcal{E}_{\text{test}}|, \overline{|\mathcal{D}_{\text{test}}|})$. We repeat this whole procedure for different $\mu_{acc} \in \{0.3, 0.35, \ldots, 0.95\}$ to cover a wide range of possible model performances observed across many datasets (see Table 3).

Figure 1 shows CI coverage probability and width for many $(C, |\mathcal{E}_{\text{test}}|, \overline{|\mathcal{D}_{\text{test}}|})$ configurations. First, we find in Figure 1a that sufficiently-many test episodes (i.e., $|\mathcal{E}_{\text{test}}| > 60$) is needed to guarantee coverage probability of our CIs is within one percentage point of the target 95%, a trend that holds regardless of compute budget. Small $|\mathcal{E}_{\text{test}}|$ also corresponds to large CI widths across all considered budgets in Figure 1b. This suggests that the choices of $|\mathcal{E}_{\text{test}}| = 1, 5, 10$ in prior work [4, 24, 56, 75] can mean inaccurate and wide CIs, while choices of $|\mathcal{E}_{\text{test}}| = 1000$ [5] can be prohibitively costly for methods with high training cost.

Next, Figure 1b reveals (i) diminishing returns in CI width (decrease in $y$-axis) as compute increases, and (ii) existence of an optimal balance between $|\mathcal{E}_{\text{test}}|$ and $\overline{|\mathcal{D}_{\text{test}}|}$ for each budget. Restricting our consideration to budgets with optima satisfying sufficient coverage probability ($|\mathcal{E}_{\text{test}}| > 60$), the minimum viable budget is 36 GPU-hours. Then, assessing the marginal benefit of each 12 GPU-hour budget increase in terms of marginal reduction in CI width between optima, we arrive at our FLEX

---

[16]Costs estimated using a Quadro RTX-8000 GPU with 48Gb memory. For few-shot settings, model was trained with 300 steps. Per-episode and per-test-example costs were approx. 95–98 and 0.7–0.11 GPU-sec, respectively. Using a model with high per-episode cost for this analysis allows us to define a lower-bound sample size requirement; we can always test inexpensive or zero-shot models on more $|\mathcal{E}_{\text{test}}|$ or $\overline{\mathcal{D}_{\text{test}}}$ within budget.

configuration of $|\mathcal{E}_{\text{test}}| = 90$ and $\overline{|\mathcal{D}_{\text{test}}|} \approx 470$ under a budget of $C = 48$ GPU-hours.[17] Further details are in Appendix B.

## 6   UniFew: A Few-Shot Learning Model by Unifying Pre-training and Downstream Task Formats

Despite their encouraging results, existing works on few-shot learning in NLP are based on either customized and often complex meta-learning algorithms [3, 4, 5, 60], heavy manual/automated engineering of textual descriptions or prompts [24, 55, 59, 78], ordering of training examples [44, 56], extensive hyperparameter tuning on held-out sets [24, 44, 55], or custom learning algorithms [55, 65]. We present UniFew, a strong few-shot learning model across *all* transfer settings and datasets tested, that eschews the need for incorporating the above-mentioned complexities and challenges.

UniFew is a prompt-based model [56], a class of models that tailor the input/output format of their data to match the format used during pretraining. While this technique allows them to perform a task without the need for additional classification layers, prompt-based models are typically sensitive to the choice of the prompts, which can require extensive search, trial-and-error, and even additional models to get right [24, 78]. To avoid this issue while still leveraging the strong capabilities of pretrained models, UniFew (1) converts examples into multiple-choice question-answer (QA) format, and (2) uses UnifiedQA [34], a T5 [51] model further pretrained on a large collection of QA pairs.[18,19]

Compared to other prompt-based models, UniFew has two main strengths. First, the prompt design problem is much simpler because UnifiedQA questions had well-defined formats. For example, we only need four general prompt templates which cover all 20 datasets in the FLEX benchmark, while prior works have needed specialized prompts for each dataset. Second, UnifiedQA's multiple-choice format ensures the model outputs a valid class label, without the need for learned or manually-defined mappings or verbalizers required for other prompt-based methods [24, 55].[20] In concurrent work, Zhong et al. [80] also show the benefit of performing meta-tuning on a variety of datasets; while their task setup as Q/A is similar to UniFew, they focus exclusively on binary zero-shot classification tasks and, unlike UniFew, do not handle multi-class or few-shot problems.

We experiment with UniFew both without and with meta-training on the FLEX benchmark's meta-training data, following the FLEX protocol (§4.2). We call the meta-trained variant UniFew$_{\text{meta}}$. We use simple prompts in the format of question followed by choices followed by the answer (according to the UnifiedQA original format). The exact prompts used are provided in Appendix C.

**Training details**   For meta-training and meta-validation of UniFew, we sampled $\mathcal{E}_{\text{train}}$ and $\mathcal{E}_{\text{val}}$ with 5-class, 5-training-shot sampling with the same number of shots per class.[21] We trained the model for total number of 30K steps, using a linear learning rate scheduler with peak rate of $3e{-}5$, 200 warmup steps, and batch size of 4; we selected the best checkpoint based on $\mathcal{E}_{\text{val}}$ performance. At meta-test time, for each episode, we trained the model on the episode's training examples (if they exist) and predicted the outputs on test examples. For training at meta-test time, we used constant learning rate of $3e{-}5$ and batch size of 4, and trained the model for 400 steps.[22] We used NVidia RTX8000 GPUs, which take about 7 GPU-hours for meta-training and 48 GPU-hours for meta-testing. For meta-testing we split the episodes among 8 GPUs to speed up evaluations.

## 7   Experiments

**Comparing UniFew with prior work**   To demonstrate the efficacy of UniFew, we evaluate it against state-of-the-art approaches for few-shot and meta-learning in NLP: LM-BFF [24], a language

---

[17]Consider budget increases $36 \to 48$, $48 \to 60$, $60 \to 72$ and $72 \to 80$. The first reduces CI width by 13%. Further increases reduce CI width by an additional 9%, 7%, and 5%, respectively. We choose $C = 48$ based on these diminishing returns.

[18]UnifiedQA and T5 both use Apache License 2.0. We use publicly-released large-size model weights.

[19]None of the supervised datasets in the pretraining of UnifiedQA or T5 are in FLEX.

[20]In rare cases, especially for zero-shot, UnifiedQA may generate an invalid answer (e.g., "Yes, Yes, No" instead of "Yes"). We use simple heuristics to normalize the answer in such cases.

[21]Users of FLEX can specify the sampling configuration of $\mathcal{E}_{\text{train}}$ and $\mathcal{E}_{\text{val}}$ as desired.

[22]For comparison with [24] we trained the model for 600 steps.

Table 2: Comparing UniFew with prior methods on their respective test suites, reporting mean accuracy (and standard deviation). For each test suite, for each result set on same number of shots, we indicate with ▷ when results are directly comparable: (i) either both use meta-training (H-SMLMT & DS with UniFew$_{meta}$) or neither do (LM-BFF with UniFew). We **bold** the better of the two.

(a) H-SMLMT (Bansal et al. [4])

| | Model | Shots | CNLL | SciT |
|---|---|---|---|---|
| ▷ | H-SMLMT | 4 | 57.6 $\pm7.1$ | 76.8 $\pm3.4$ |
| | UniFew | 4 | 76.6 $\pm2.6$ | 65.1 $\pm9.9$ |
| ▷ | UniFew$_{meta}$ | 4 | **79.7** $\pm2.8$ | **85.4** $\pm2.5$ |
| ▷ | H-SMLMT | 8 | 70.2 $\pm3.0$ | 79.1 $\pm1.1$ |
| | UniFew | 8 | 80.6 $\pm3.7$ | 70.9 $\pm5.2$ |
| ▷ | UniFew$_{meta}$ | 8 | **81.2** $\pm3.8$ | **86.8** $\pm1.4$ |
| ▷ | H-SMLMT | 16 | 80.6 $\pm2.8$ | 80.4 $\pm1.4$ |
| | UniFew | 16 | 85.8 $\pm1.9$ | 76.7 $\pm4.6$ |
| ▷ | UniFew$_{meta}$ | 16 | **87.9** $\pm1.9$ | **85.4** $\pm2.5$ |

(b) LM-BFF (Gao et al. [24])

| | Model | Shots | CR | MR | SNLI | Subj | TREC |
|---|---|---|---|---|---|---|---|
| ▷ | LM-BFF$_{man}$[23] | 0[24] | **79.5** | **80.8** | 49.5 | **51.4** | **32.0** |
| ▷ | UniFew | 0 | 78.8 | 74.8 | **54.4** | 50.3 | 15.0 |
| | UniFew$_{meta}$ | 0 | 92.1 | 90.5 | 83.8 | 56.8 | 39.1 |
| ▷ | LM-BFF | 16/16[25] | 91.0 $\pm0.9$ | **87.7** $\pm1.4$ | **77.5** $\pm3.5$ | **91.4** $\pm1.8$ | **89.4** $\pm1.7$ |
| ▷ | UniFew | 16/16 | **92.2** $\pm0.8$ | 87.2 $\pm0.1$ | 75.6 $\pm1.5$ | 84.6 $\pm5.4$ | 86.7 $\pm0.3$ |
| | UniFew$_{meta}$ | 16/16 | 92.7 $\pm0.4$ | 90.2 $\pm0.8$ | 84.9 $\pm0.5$ | 87.6 $\pm2.0$ | 86.1 $\pm0.4$ |

(c) Distributional Signature (Bao et al. [5])

| | Model | Shots | Amzn[†] | FRel[†] | HuffP[†] | 20N[†] | Reut[†] |
|---|---|---|---|---|---|---|---|
| ▷ | DS | 1 | 62.7 $\pm0.7$ | 67.1 $\pm0.9$ | 43.1 $\pm0.2$ | 52.2 $\pm0.7$ | 81.8 $\pm1.6$ |
| | UniFew | 1 | 82.1 $\pm8.5$ | 75.7 $\pm13.2$ | 65.9 $\pm13.4$ | 58.4 $\pm11.6$ | 92.0 $\pm8.3$ |
| ▷ | UniFew$_{meta}$ | 1 | **84.3** $\pm8.9$ | **90.6** $\pm6.2$ | **78.6** $\pm6.9$ | **70.3** $\pm9.1$ | **96.9** $\pm2.5$ |
| ▷ | DS | 5 | 81.2 $\pm0.3$ | 83.5 $\pm0.3$ | 63.5 $\pm0.1$ | 68.3 $\pm0.2$ | 96.0 $\pm0.3$ |
| | UniFew | 5 | 88.5 $\pm7.4$ | 88.8 $\pm6.5$ | 77.1 $\pm6.0$ | 72.2 $\pm8.4$ | 97.0 $\pm2.8$ |
| ▷ | UniFew$_{meta}$ | 5 | **90.5** $\pm5.9$ | **93.1** $\pm4.4$ | **81.7** $\pm5.2$ | **76.2** $\pm7.1$ | **98.0** $\pm2.0$ |

model prompt-based fine-tuning method, as well as Distributional Signatures (DS) [5] and H-SMLMT [4], two state-of-the-art meta-learning techniques. Refer to Appendix D for details on these methods.

We compare to these methods using the datasets in the FLEX benchmark to establish the quality of our model. Since we constructed our benchmark from disjoint subsets of datasets evaluated in each of these prior works (§4.1), we compare each method with its corresponding subset of datasets. Each of these prior works evaluates their methods using different experimental setups (classes, number of episodes, shots) than our benchmark and was not designed to handle FLEX's challenging episode characteristics like class imbalance. To enable fair comparison, we test UniFew on the exact data splits released by the authors when available (H-SMLMT and LM-BFF). For DS, we sample (balanced) episodes using our framework after matching their test settings (number of shots and classes, class splits, etc.) and reproduce their reported results to within 1% absolute difference using their model code; we use these episodes for our experiments. The results in Table 2 show that UniFew$_{meta}$ outperforms both H-SMLMT and DS meta-learning approaches by relatively large margins, while achieving competitive results compared with LM-BFF. Note that UniFew's strong results are without meta-learning approaches, extensive prompt-engineering, or per-episode hyperparameter search.

**Evaluating UniFew on the FLEX benchmark** Having established UniFew as a strong model comparable to recent, state-of-the art techniques, we present its results on the final version of our benchmark (with class imbalance, etc.). From Table 3, we observe three findings. First, pretraining is an effective technique for infusing an NLP model with the ability to perform few-shot generalization even without any meta-training, as UniFew is able to score $\Delta_{few} = +12.8$ higher when provided

---

[23]Gao et al. [24]'s automatic prompt search and in-context learning are not available in the zero-shot setting, so they instead use manually-designed prompts.

[24]Zero-shot results from Gao et al. [24] are on the entire test set, so there is no reported standard deviation.

[25]16/16 denotes 16 shots for training plus 16 more for validation which we only use for early stopping while Gao et al. [24] use for grid-search hyperparameter tuning.

Table 3: Mean accuracy of UniFew and UniFew_meta on FLEX benchmark in zero and few-shot setups.

| | Zero-shot | | | | Few-shot | | | | |
| --- | --- | --- | --- | --- | --- | --- | --- | --- | --- |
| | Class | Domain | Task | Overall | Class | Domain | Task | Overall | $\Delta_{few}$ (Overall) |
| UniFew | 59.5 | 67.9 | 36.6 | 56.5 | 75.8 | 72.4 | 54.3 | 69.3 | +12.8 |
| UniFew_meta | 75.6 | 87.6 | 41.1 | 71.0 | 80.2 | 86.8 | 62.4 | 77.9 | +6.9 |
| $\Delta_{meta}$ | +16.2 | +19.7 | +4.5 | +14.5 | +4.3 | +14.4 | +8.1 | +8.6 | |

few rather than zero examples. Second, by comparing UniFew_meta and UniFew, we see that meta-training has a substantial impact on zero-shot performance ($\Delta_{meta} = +14.5$), but its benefit, while still substantial, is less in the few-shot setting ($\Delta_{meta} = +8.6$). Third, while meta-training adds roughly the same benefit to zero and few-shot performance for both domain and task transfer settings, meta-training disproportionately benefits zero-shot class transfer ($\Delta_{meta} = +16.2$) over few-shot class transfer ($\Delta_{meta} = +4.3$). Such observations are made possible through unified evaluation and comparison across different transfer types. The full FLEX benchmark results broken down by individual datasets are in Appendix E.

## 8 Limitations and Future Work

While the initial FLEX benchmark is focused on classification tasks, we aim to use our benchmark creation toolkit (§4.4) to incorporate additional task formats like span selection or text generation. Furthermore, the benchmark currently only supports English language tasks; to study *language transfer*, we aim to incorporate new datasets using our toolkit. Adding diverse datasets has its own challenges; while we've selected datasets for our benchmark based on prior work adoption and have attempted to verify their licensing for research use, we were unable to find license details for some datasets (Appendix A). We believe it is crucial to continually evolve the suite of datasets to remain challenging for the best models [36] and to tackle real-world challenges [1].

In addition, Sample Size Design (§5) simulations currently rely on our own available training estimates. We plan to gather a more representative sample from community leaderboard submissions.

Our public leaderboard could benefit from extended support for detailed comparisons between submissions based on properties of techniques. For example, approaches may vary in terms of model characteristics (e.g., number of parameters), data and supervision used during pretraining, amount of compute, etc. We encourage reporting all these factors to enable the community to analyze and make progress on important sub-spaces in the overall few-shot technique design space.

Finally, we believe the benefits of improving few-shot NLP techniques outweigh potential risks, but we acknowledge potential harms associated with language models [7, 14, 57, 63]. Few-shot models learn a task from a few examples but rely heavily on knowledge encoded in the pretrained model. Thus, few-shot models are more likely to inherit the biases of the pretrained models, compared to more fully supervised models; as the community focuses more on few-shot learning, it is more important than ever for future pretrained models to be careful about biases in the underlying pretraining corpora.

## 9 Conclusion

In this work, we unify and bring rigor to few-shot NLP evaluation. We formulate the FLEX Principles, a set of requirements and best practices that enables unified, rigorous, valid, and cost-sensitive measurement. We advance the principles with new Sample Size Design methodology for optimizing statistical accuracy and precision while keeping costs low. The FLEX benchmark is our instantiation of the FLEX Principles; it employs Sample Size Design and includes four few-shot transfer settings, zero-shot evaluation, and a public leaderboard with diverse NLP tasks. We present UniFew, a prompt-based model that aligns pretraining and downstream task formats, achieving results competitive with recent few-shot methods despite using trivial prompt engineering. Finally, we release an extensible, open-source toolkit (used to train UniFew and generate the FLEX benchmark) to support future benchmark creation and few-shot NLP model training.

## Acknowledgments and Disclosure of Funding

We would like to thank Chandra Bhagavatula, Matt Gardner, Matt Peters, Doug Downey, Dan Weld, and the four anonymous reviewers for helpful comments, suggestions and feedback. We would also like to acknowledge the large community effort involved in the creation of the datasets and open-source tools we utilize.

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
