## A Datasets

**Dataset stats, tasks, and transfer types**   Table 4 summarizes the tasks and datasets used for meta-training and meta-testing. To enable automated benchmark construction and maximize access, we restrict datasets to those that are freely available for automated download.[26] We include the GLUE tasks used by Bansal et al. [3, 4] for meta-training[27] and thus exclude GLUE tasks used by Gao et al. [24] from meta-testing. Although Bansal et al. [3, 4] additionally use SNLI for meta-training, we reserve it for meta-testing for comparison to Gao et al. [24] and because NLI is already represented in the meta-training datasets.

**Textual labels and licenses for datasets**   We made CoNLL labels more descriptive from their original PER,ORG,LOC,MISC. For TREC, we used the more readable labels from the manual template in [24]. For readability, Amazon labels are shown without underscores and Amazon and HuffPost capitalization has been removed. License information is shown in parentheses.

- MR [49] (license unavailable[28]): **Test:** negative, positive
- CR [32] (license unavailable[29]): **Test:** negative, positive
- Subj [48] (license unavailable[30]): **Test:** objective, subjective
- TREC [69] (license unavailable[31]): **Test:** description, entity, expression, human, location, number
- FewRel [28] (MIT License[32]): **Train:** applies to jurisdiction, architect, child, competition class, constellation, contains administrative territorial entity, country, country of citizenship, country of origin, crosses, father, field of work, followed by, follows, genre, has part, head of government, headquarters location, heritage designation, instance of, instrument, league, licensed to broadcast to, located in or next to body of water, located in the administrative territorial entity, located on terrain feature, location, location of formation, manufacturer, member of, member of political party, military branch, military rank, mother, mountain range, mouth of the watercourse, movement, notable work, occupant, occupation, operating system, operator, owned by, part of, participant, participant of, participating team, place served by transport hub, position held, position played on team / speciality, record label, religion, residence, said to be the same as, sibling, sport, sports season of league or competition, spouse, subsidiary, successful candidate, taxon rank, tributary, voice type, winner, work location; **Val:** developer, director, original network, performer, publisher; **Test:** after a work by, characters, composer, distributor, language of work or name, main subject, nominated for, original language of film or TV show, platform, screenwriter
- HuffPost [46] (CC0: Public Domain[33]): **Train:** arts, arts & culture, black voices, comedy, culture & arts, fifty, food & drink, good news, green, impact, latino voices, media, money, parenting, religion, sports, style, the worldpost, travel, women; **Val:** crime, queer voices, science, weird news, worldpost; **Test:** business, college, divorce, education, entertainment, environment, healthy living, home & living, parents, politics, style & beauty, taste, tech, weddings, wellness, world new
- CoNLL [66] (license for research by Reuters[34]): **Test:** location, organization, other, person
- SNLI [9] (Creative Commons Attribution-ShareAlike 4.0 International License[35]): **Test:** contradiction, entailment, neutral

---

[26]We exclude RCV1 (used by [5]) and MPQA (used by [24]), since they require agreeing to license terms through web forms at download time.

[27]We follow Bansal et al. [3] and use the matched+mismatched version of MNLI and exclude WNLI and STS-B from meta-training due to the small training size and regression task format, respectively

[28]https://www.cs.cornell.edu/people/pabo/movie-review-data/rt-polaritydata.README.1.0.txt

[29]https://www.cs.uic.edu/~liub/FBS/CustomerReviewData.zip

[30]https://www.cs.cornell.edu/people/pabo/movie-review-data/subjdata.README.1.0.txt

[31]https://cogcomp.seas.upenn.edu/Data/QA/QC/

[32]https://huggingface.co/datasets/few_rel

[33]https://www.kaggle.com/rmisra/news-category-dataset

[34]https://www.clips.uantwerpen.be/conll2003/ner/

[35]https://huggingface.co/datasets/snli

- SciTail [35] (license unavailable[36]): **Test:** entailment, neutral
- Amazon [30] (license unavailable[37]): **Train:** automotive, baby, beauty, cell phones and accessories, grocery and gourmet food, health and personal care, home and kitchen, patio lawn and garden, pet supplies, sports and outdoor; **Val:** apps for android, cds and vinyl, digital music, toys and games, video games; **Test:** amazon instant video, books, clothing shoes and jewelry, electronics, kindle store, movies and tv, musical instruments, office products, tools and home improvement
- 20News [38] (license unavailable[38]): **Train:** rec.autos, rec.motorcycles, rec.sport.baseball, rec.sport.hockey, sci.crypt, sci.electronics, sci.med, sci.space; **Val:** comp.graphics, comp.os.ms-windows.misc, comp.sys.ibm.pc.hardware, comp.sys.mac.hardware, comp.windows.x; **Test:** alt.atheism, misc.forsale, soc.religion.christian, talk.politics.guns, talk.politics.mideast, talk.politics.misc, talk.religion.misc
- Reuters [41] (license for research by Reuters[39]): **Train:** acq, alum, bop, cocoa, coffee, copper, cotton, cpi, crude, earn, gnp, gold, grain, interest, ipi; **Val:** iron-steel, jobs, livestock, money-fx, money-supply; **Test:** nat-gas, orange, reserves, retail, rubber, ship, sugar, tin, trade, veg-oil, wpi
- CoLa [72] (released under fair use[40]): **Val:** acceptable, unacceptable
- MNLI [74] (multiple licenses[41]): **Train/Val:** contradiction, entailment, neutral
- MRPC [20] (license unavailable[42]): **Train/Val:** equivalent, not_equivalent
- QNLI [52] (CC BY-SA 4.0[43]): **Train/Val:** entailment, not_entailment
- QQP [70] (non-commercial use[44]): **Train/Val:** duplicate, not_duplicate
- RTE [6, 8, 17, 26] (license unavailable[45]): **Train/Val:** entailment, not_entailment
- SST-2 [62] (license unavailable[46]): **Train/Val:** negative, positive
- WNLI [40] (CC BY 4.0[47]): **Val:** entailment, not_entailment

# B   Sample Size Simulations

We describe how we performed the simulations described in §5.

**Relating $C$, $|\mathcal{E}_{\textbf{test}}|$ and $\overline{|\mathcal{D}_{\textbf{test}}|}$**    The cost of meta-testing on FLEX for a given dataset is the sum of the cost of both few-shot and zero-shot evaluations:

$$
\begin{aligned}
C &= C_{\text{few}} + C_{\text{zero}} \\
&= \left( C_{\text{few}}^{E}|\mathcal{E}_{\text{test}}| + C_{\text{few}}^{I}|\mathcal{E}_{\text{test}}|\overline{|\mathcal{D}_{\text{test}}|} \right) + \left( C_{\text{zero}}^{E}|\mathcal{E}_{\text{test}}| + C_{\text{zero}}^{I}|\mathcal{E}_{\text{test}}|\overline{|\mathcal{D}_{\text{test}}|} \right) \\
&= |\mathcal{E}_{\text{test}}| \left( (C_{\text{few}}^{E} + C_{\text{zero}}^{E}) + \overline{|\mathcal{D}_{\text{test}}|}(C_{\text{few}}^{I} + C_{\text{zero}}^{I}) \right)
\end{aligned}
$$

where $C_{\text{few|zero}}^{E}$ is (average) time spent per-episode during model setup and training, $C_{\text{few|zero}}^{I}$ is (average) time spent per-episode per-test-instance on evaluation. We estimate these quantities on a

---

[36]https://allenai.org/data/scitail
[37]http://jmcauley.ucsd.edu/data/amazon/
[38]https://huggingface.co/datasets/newsgroup
[39]https://kdd.ics.uci.edu/databases/reuters21578/README.txt
[40]https://nyu-mll.github.io/CoLA/
[41]https://www.aclweb.org/anthology/N18-1101.pdf
[42]https://www.microsoft.com/en-us/download/details.aspx?id=52398
[43]https://rajpurkar.github.io/SQuAD-explorer/
[44]https://www.kaggle.com/quora/question-pairs-dataset
[45]https://gluebenchmark.com/
[46]https://nlp.stanford.edu/sentiment/
[47]https://cs.nyu.edu/~davise/papers/WinogradSchemas/WS.html

Table 4: FLEX datasets. Use in prior few-shot evaluation marked indicated with * [24], † [3], and ‡ [5]. $|\mathcal{Y}_{\text{val}}| = (k)$ parentheses indicate that the same classes are reused between training and validation. The notation $\{i{:}j\}$ is used to denote the set of all integers between $i$ and $j$, inclusive. "class." and "doc." are shorthand for "classification" and document". The "–" indicates that the corresponding dataset is not used for a certain phase, for example, CoLa and WNLI are only used for meta-validation.

| Task Type | Dataset | $|\mathcal{Y}_{\text{train}}|$ | $|\mathcal{Y}_{\text{val}}|$ | $|\mathcal{Y}_{\text{test}}|$ | $|\mathcal{Y}_{\text{test}}|$/ep. | #test ex. | Transfer |
|---|---|---|---|---|---|---|---|
| | | | Single-sentence tasks | | | | |
| sentiment | MR* | – | – | 2 | {2} | 10662 | Domain & Pretrain |
| sentiment | CR* | – | – | 2 | {2} | 1708 | Domain & Pretrain |
| subjectivity | Subj* | – | – | 2 | {2} | 10000 | Task & Pretrain |
| question class. | TREC* | – | – | 6 | {6} | 500 | Task & Pretrain |
| entity typing | CoNLL† | – | – | 4 | {4} | 5648 | Task & Pretrain |
| relation class. | FewRel‡ | 65 | 5 | 10 | {5:10} | 7000 | Class & Pretrain |
| news headline topic | HuffPost‡ | 20 | 5 | 16 | {5:10} | 113957 | Class & Pretrain |
| sentiment | SST-2 † | 2 | (2) | – | – | – | – |
| acceptability | CoLa† | – | 2 | – | – | – | Task & Pretrain |
| | | | Sentence-pair tasks | | | | |
| NLI | SNLI* | – | – | 3 | {3} | 9842 | Domain & Pretrain |
| NLI | SciTail† | – | – | 2 | {2} | 2126 | Domain & Pretrain |
| NLI | MNLI† | 3 | (3) | – | – | – | – |
| QA/NLI | QNLI† | 2 | (2) | – | – | – | – |
| NLI | RTE† | 2 | (2) | – | – | – | – |
| paraphrase | MRPC† | 2 | (2) | – | – | – | – |
| paraphrase | QQP† | 2 | (2) | – | – | – | – |
| NLI | WNLI | – | 2 | – | – | – | Domain & Pretrain |
| | | | Document tasks | | | | |
| review product | Amazon‡ | 10 | 5 | 9 | {5:9} | 9000 | Class & Pretrain |
| informal doc. topic | 20News‡ | 8 | 5 | 7 | {5:7} | 6021 | Class & Pretrain |
| document topic | Reuters‡ | 15 | 5 | 11 | {5:10} | 835 | Class & Pretrain |

single Titan RTX-8000 GPU with 48Gb memory by conducting meta-testing runs with the UniFew model (300 steps in few-shot setting) across all datasets in FLEX with arbitrary choices for $|\mathcal{D}_{\text{test}}^E|$. These tended to be around 95–98 sec for $C_{\text{few}}^E$, 1–3 sec for $C_{\text{zero}}^E$, and 0.7–0.11 sec for $C_{\text{few|zero}}^I$. From this, we derived possible $(C, |\mathcal{E}_{\text{test}}|, \overline{|\mathcal{D}_{\text{test}}|})$ configurations by solving for $\overline{|\mathcal{D}_{\text{test}}|}$ over grids of $C = 24, 36, \ldots, 84$ and $|\mathcal{E}_{\text{test}}| = 5, 15, 30, 45, \ldots, 150$.

**Simulating confidence intervals**   We describe a single simulation run by a given $(C, |\mathcal{E}_{\text{test}}|, \overline{|\mathcal{D}_{\text{test}}|})$. First, we need to generate $\overline{|\mathcal{D}_{\text{test}}|}$ model predictions for every episode $E \in \mathcal{E}_{\text{test}}$. To do this, we assume each episode has a latent episode-specific model accuracy $\mu_{acc}^{(1)}, \ldots, \mu_{acc}^{(|\mathcal{E}_{\text{test}}|)}$, where each $\mu_{acc}^{(\cdot)}$ is drawn from a Normal distribution with mean $\mu_{acc}$ and variance $\sigma_{acc}^2$. Here, $\mu_{acc}$ represents the unknown overall model accuracy that is our target of estimation, and $\sigma_{acc}^2$ represents inherent variability in task difficulty across episodes (e.g., due to different number of classes or imbalance). In our simulations, we set $\sigma_{acc} = 0.05$. For each episode $E$, we generate prediction outcomes (i.e. correct or incorrect) from a Bernoulli with success probability $\mu_{acc}^E$. This allows us to compute episode-specific accuracy estimates $\hat{\mu}_{acc}^{(1)}, \ldots, \hat{\mu}_{acc}^{(|\mathcal{E}_{\text{test}}|)}$ and finally compute the mean, standard deviation, and (bootstrap) CI across these episodes. In doing so, a single simulation run represents a possible submission outcome to FLEX for a given model, and we can obtain the resulting CI's width and verify whether it contains the true model accuracy $\mu_{acc}$.

## C  Prompts

We use the following prompts for FLEX benchmark tasks based on the input type:

- Single text classification:

  `Topic?\\n (A) Class1 (B) Class2 (C) Class3 \\n` *`The document`*

- Sentence-pair classification:

  *`Sentence 1`* `Is` *`Sentence 2`*`?\\n (A) Yes (B) No (C) Maybe`

- Relation classification:

  *`mention-1`* `to` *`mention-2`*`? \\n (A) Class1 (B) Class2 (C) Class3 \\n` *`Some text`* *`#mention-1# some text *mention-2* some text.`*

- Entity recognition:

  *`What is the type of the entity between the # marks?`* `\\n (A) Class1 (B) Class2 (C) Class3 \\n` *`Some text #mention-1# some text.`*

The format of question, followed by the document followed by answer choices, as well as the use of the special delimiter of \\n is according to UnifiedQA's original pretraining. We follow [24]'s format of NLI for sentence pair tasks and T5 [51] for relation classification.

## D  Baseline Models

This section briefly describes the baselines we use for comparison.

*LM-BFF [24]* is a language model prompt-based fine-tuning method with extensive automated and manual approaches for prompt generation. It also uses a strategy for dynamically and selectively incorporating demonstrations into each context which is an extension to GPT-3's in-context learning technique [10].

*Distributional Signatures (DS) [5]* A meta-learning method designed for class transfer. DS uses lexical "distributional signatures," characteristics of the underlying word distributions to transfer attention patterns across tasks within a meta-learning framework.

*SMLMT [4]* A self-supervised approach for domain and task transfer. SMLMT creates the target task distribution from a large set of unlabeled sentences used within a meta-learning framework for optimal transfer. We compare with the strongest model variant in this paper, Hybrid-SMLMT which is trained on both self-supervised and supervised tasks.

## E  Full Results Statistics

Here, we provide full results broken down by dataset in Table 5. We report bootstrap CIs and standard deviations, as recommended by Dhillon et al. [19].

## F  Software Licenses

Our code is licensed under Apache License 2.0. Our framework dependencies are:

- HuggingFace Datasets[48] (Apache 2.0)
- Hydra[49] (MIT License)
- Numpy[50] (BSD 3-Clause "New" or "Revised")
- Scipy[51] (BSD 3-Clause "New" or "Revised")
- Pandas[52] (BSD 3-Clause "New" or "Revised")

---

[48] https://github.com/huggingface/datasets/blob/master/LICENSE
[49] https://github.com/facebookresearch/hydra/blob/master/LICENSE
[50] https://github.com/numpy/numpy/blob/main/LICENSE.txt
[51] https://github.com/scipy/scipy/blob/master/LICENSE.txt
[52] https://github.com/pandas-dev/pandas/blob/master/LICENSE

Table 5: Full results table with all the stats. CI-low and CI-up are the lower and upper 95% bootstrap confidence intervals (of the mean), and CI-sem is the symmetric 95% standard error-based confidence interval.

| Shot | Model | Stat | Class Transfer | | | | | Domain Transfer | | | | Task Transfer | | |
|---|---|---|---|---|---|---|---|---|---|---|---|---|---|---|
| | | | Amzn | FRel | HufP | 20N | Reut | CR | MR | SciT | SNLI | CNLL | Subj | TREC |
| Zero | UniFew | Mean | 69.9 | 52.5 | 46.9 | 43.7 | 84.3 | 85.4 | 77.1 | 56.4 | 52.6 | 31.6 | 55.2 | 23.1 |
| | | Stdev | 7.2 | 9.7 | 10.1 | 10.2 | 6.0 | 4.7 | 3.3 | 3.2 | 2.5 | 3.3 | 3.3 | 4.7 |
| | | CI-low | 1.45 | 2.02 | 2.05 | 1.95 | 1.33 | 0.95 | 0.67 | 0.62 | 0.53 | 0.68 | 0.66 | 0.96 |
| | | CI-up | 1.50 | 1.85 | 2.15 | 2.13 | 1.24 | 0.92 | 0.64 | 0.62 | 0.51 | 0.68 | 0.68 | 0.95 |
| | | CI-sem | 1.50 | 2.02 | 2.10 | 2.12 | 1.26 | 0.99 | 0.69 | 0.66 | 0.52 | 0.69 | 0.68 | 0.97 |
| Zero | UniFew$_{meta}$ | Mean | 75.6 | 79.4 | 68.5 | 60.0 | 94.6 | 93.7 | 90.8 | 82.6 | 83.3 | 34.8 | 52.7 | 35.9 |
| | | Stdev | 8.4 | 9.2 | 6.6 | 8.3 | 2.1 | 1.3 | 1.9 | 2.2 | 2.0 | 1.8 | 2.9 | 2.0 |
| | | CI-low | 1.77 | 1.89 | 1.26 | 1.81 | 0.46 | 0.27 | 0.41 | 0.44 | 0.39 | 0.35 | 0.60 | 0.41 |
| | | CI-up | 1.81 | 1.80 | 1.36 | 1.76 | 0.45 | 0.26 | 0.41 | 0.44 | 0.40 | 0.40 | 0.58 | 0.43 |
| | | CI-sem | 1.74 | 1.91 | 1.37 | 1.72 | 0.44 | 0.27 | 0.40 | 0.46 | 0.41 | 0.38 | 0.60 | 0.42 |
| Few | UniFew | Mean | 79.5 | 79.2 | 62.8 | 63.1 | 94.5 | 90.1 | 78.6 | 64.9 | 55.8 | 44.3 | 60.5 | 58.1 |
| | | Stdev | 7.5 | 7.5 | 7.7 | 7.8 | 3.2 | 6.6 | 10.5 | 8.9 | 9.5 | 7.9 | 9.9 | 7.7 |
| | | CI-low | 1.52 | 1.54 | 1.60 | 1.59 | 0.67 | 1.42 | 2.16 | 1.67 | 2.12 | 1.76 | 1.99 | 1.71 |
| | | CI-up | 1.57 | 1.49 | 1.55 | 1.59 | 0.63 | 1.26 | 1.99 | 1.81 | 1.75 | 1.68 | 2.21 | 1.63 |
| | | CI-sem | 1.56 | 1.55 | 1.59 | 1.62 | 0.66 | 1.37 | 2.18 | 1.85 | 1.97 | 1.65 | 2.06 | 1.61 |
| Few | UniFew$_{meta}$ | Mean | 82.1 | 87.2 | 67.9 | 67.3 | 96.3 | 93.2 | 89.4 | 83.6 | 80.9 | 58.6 | 68.7 | 60.0 |
| | | Stdev | 7.0 | 5.7 | 7.5 | 7.8 | 2.5 | 2.5 | 2.8 | 4.7 | 4.5 | 4.7 | 10.6 | 6.6 |
| | | CI-low | 1.44 | 1.20 | 1.62 | 1.58 | 0.55 | 0.55 | 0.59 | 1.00 | 0.95 | 1.04 | 2.30 | 1.41 |
| | | CI-up | 1.53 | 1.07 | 1.49 | 1.58 | 0.52 | 0.48 | 0.58 | 0.92 | 0.89 | 0.93 | 2.24 | 1.39 |
| | | CI-sem | 1.46 | 1.19 | 1.56 | 1.61 | 0.53 | 0.52 | 0.58 | 0.98 | 0.93 | 0.97 | 2.19 | 1.38 |

Table 6: Mean accuracy (with 95% standard error-based CIs) of UniFew and UniFew$_{meta}$ on FLEX benchmark in zero and few-shot settings.

| Shot | Model | Class Transfer | | | | | Domain Transfer | | | | Task Transfer | | | Avg |
|---|---|---|---|---|---|---|---|---|---|---|---|---|---|---|
| | | Amzn | FRel | HufP | 20N | Reut | CR | MR | SciT | SNLI | CNLL | Subj | TREC | |
| Zero | UniFew | 69.9 ±7.2 | 52.5 ±9.7 | 46.9 ±10.1 | 43.7 ±10.2 | 84.3 ±6.0 | 85.4 ±4.7 | 77.1 ±3.3 | 56.4 ±3.2 | 52.6 ±2.5 | 31.6 ±3.3 | 55.2 ±3.3 | 23.1 ±4.7 | 56.5 |
| | UniFew$_{meta}$ | 75.6 ±1.7 | 79.4 ±1.9 | 68.5 ±1.4 | 60.0 ±1.7 | 94.6 ±0.4 | 93.7 ±0.3 | 90.8 ±0.4 | 82.6 ±0.5 | 83.3 ±0.4 | 34.8 ±0.4 | 52.7 ±0.6 | 35.9 ±0.4 | 71.0 |
| Few | UniFew | 79.5 ±7.5 | 79.2 ±7.5 | 62.8 ±7.7 | 63.1 ±7.8 | 94.5 ±3.2 | 90.1 ±6.6 | 78.6 ±10.5 | 64.9 ±8.9 | 55.8 ±9.5 | 44.3 ±7.9 | 60.5 ±9.9 | 58.1 ±7.7 | 69.3 |
| | UniFew$_{meta}$ | 82.1 ±1.5 | 87.2 ±1.2 | 67.9 ±1.6 | 67.3 ±1.6 | 96.3 ±0.5 | 93.2 ±0.5 | 89.4 ±0.6 | 83.6 ±1.0 | 80.9 ±0.9 | 58.6 ±1.0 | 68.7 ±2.2 | 60.0 ±1.4 | 77.9 |

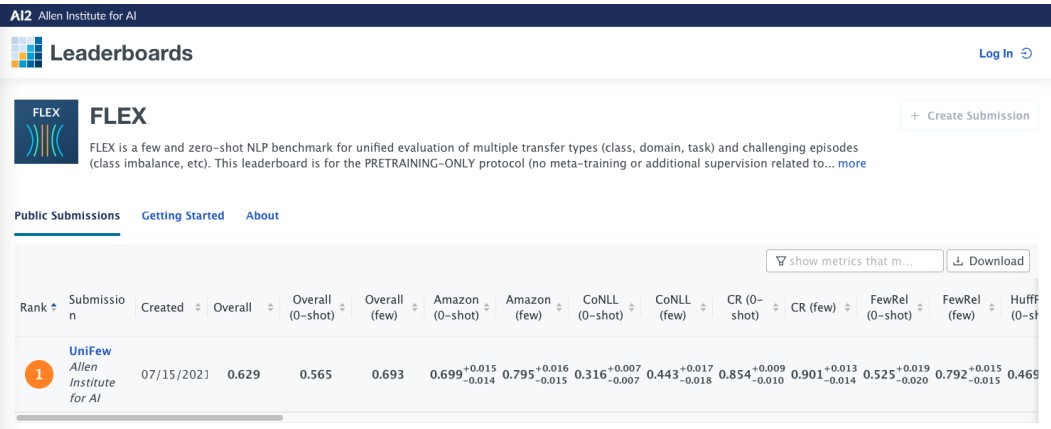

Figure 2: The FLEX public leaderboard for the Pretraining-Only evaluation protocol. A separate leaderboard exists (not shown) for the Meta-Trained protocol.

- Scikit-learn[53] (BSD 3-Clause "New" or "Revised")
- Tqdm[54] (MIT License, MPLv2.0)
- Click[55] (MIT License)

Additional dependencies used in UniFew are:

- Transformers[56] (Apache 2.0)
- PyTorch[57] (Misc)
- Pytorch Lightning[58] (Apache 2.0)

See Appendix A for dataset licenses.

## G Leaderboard

Figure 2 shows the leaderboard results interface, which displays individual dataset scores with bootstrapped confidence intervals, standard deviations, as well as macro-averaged "Overall" scores.

---

[53]https://github.com/scikit-learn/scikit-learn/blob/main/COPYING
[54]https://github.com/tqdm/tqdm/blob/master/LICENCE
[55]https://github.com/kohler/click/blob/master/LICENSE
[56]https://github.com/huggingface/transformers/blob/master/LICENSE
[57]https://github.com/pytorch/pytorch/blob/master/LICENSE
[58]https://github.com/PyTorchLightning/pytorch-lightning/blob/master/LICENSE