# OpenReview forum: "FLEX: Unifying Evaluation for Few-Shot NLP"
_NeurIPS.cc/2021/Conference — NeurIPS 2021 Poster_

### Official Review · Reviewer_d7bJ · 2021-07-14

**Rating:** 6
**Confidence:** 4

**Summary:**

This paper proposes FLEET, a benchmark for few-shot learning in NLP. FLEET is comprised of different few-shot settings (new tasks, label sets, or domains at test time), different amount of training examples ("shots"),  and meta-learning strategies (w/ and w/o meta learning). The paper goes through the design decisions behind FLEET and detail its construction. Finally, they present a baseline based on UnifiedQA, wherein they reformulate downstream tasks as QA and then finetune the model on downstream tasks.

**Limitations And Societal Impact:**

Yes.

**Main Review:**

First, the benchmark contains a lot of different settings. On one hand, this is a positive as it allows many different types of research to be conducted with the data. On the other hand, it practically makes understanding, reporting, and using the benchmark a bit messy. For example, people will report very specific settings such as FLEET with {4-shot, with meta learning, task transfer}. More generally, there is often a tradeoff when constructing datasets of "do we focus on specific problems/settings we think are interesting" versus "do we build datasets that many people could use in different settings". This paper is a very extreme on trying to satisfy the second question: rather than trying to take a stand on what the community should work on, it tries to do a bit of everything.

Second, a common criticism of GLUE and other benchmarks which applies to FLEET is that the benchmarks (1) simply repackage and resell other datasets as a new benchmark without adding substantial new value and (2) contain a relatively ad-hoc selection of individual datasets. FLEET is similar in that it sort of throws together a collection of tasks (sentiment, NLI, etc.) mainly guided by low-level reasons (e.g., the class imbalances) rather than higher-level ones (e.g., based on practical few-shot scenarios, different types of reasoning, tests of spurious correlations, etc.).

Third, a lower-level point is that I think few-shot benchmarks should contain multiple prompts / label descriptions for each dataset. There can be high variance across different prompts (Zhao et al. ICML 2021, Gao et al. ACL 2021, etc.) and so I think the reaction to this should be to report accuracy across multiple different prompts.

Finally, the proposed method is extremely similar to this concurrent work https://arxiv.org/abs/2104.04670. This is not a negative or a positive, I am simply pointing this out as an FYI to the authors, as they should probably cite/discuss this in a future version of their draft.

**Time Spent Reviewing:**

2

---

> ### Author Response · Authors · 2021-08-10
> **Response**
>
> Thank you for taking the time to review our paper and for your thoughtful comments.
>
> > On the other hand, it practically makes understanding, reporting, and using the benchmark a bit messy. For example, people will report very specific settings such as FLEET with {4-shot, with meta learning, task transfer}
>
> FLEET provides a general test of few-shot ability, e.g., sampling a variety of episodes (not just 4-shot) and reporting aggregate performance. Users are not encouraged to report specific settings like 4-shot (those would not have enough statistical accuracy). At the same time, we report a breakdown by dataset and transfer type in order to assist with diagnostics and performance analysis.
>
> > rather than trying to take a stand on what the community should work on, it tries to do a bit of everything
>
> We believe FLEET takes a strong stand on _how_ few-shot evaluation should be conducted, by incorporating best practices from domains outside NLP, providing new methodology for optimizing statistical accuracy, and unifying multiple transfer types (which previously did not compare to one another). We believe enabling rigorous comparison of methods is the first step to forming a community opinion about what is most valuable to work on; FLEET is opinionated in its design decisions and easily extensible to different mixes of datasets as techniques improve and community focus shifts.
>
> >  a common criticism of GLUE... is that the benchmarks (1) simply repackage and resell other datasets as a new benchmark without adding substantial new value and (2) contain a relatively ad-hoc selection of individual datasets
>
> We agree that simply repackaging is not useful, but FLEET provides much more. Past evaluation suites do not follow the design principles we introduce, and prior to FLEET, direct comparison of few-shot approaches for different transfer types was not possible. The inclusion of multiple transfer types and zero/few-shot enabled new kinds of analysis, e.g., our observation that meta-training disproportionately helps class transfer in the zero-shot setting (lines 308-309).
>
> > rather than higher-level ones (e.g., based on practical few-shot scenarios, different types of reasoning, tests of spurious correlations, etc.)
>
> Our design _is_ guided by many of these higher-level principles. Sampling variable shots, including class imbalance, disallowing large validation sets, etc are all practical considerations that make evaluation more realistic. Our selection of datasets was also chosen to test specific types of transfer, analogous to testing “different types of reasoning”.
>
> We agree that there are many additional questions about few-shot transfer that should be asked in the future. We carefully designed our desiderata and framework tooling to provide a strong foundation for the creation of future benchmarks, and invite researchers to use our tools to specify new meta-training/testing splits that focus on other phenomena of interest that emerge, spurious correlations, etc.
>
> > few-shot benchmarks should contain multiple prompts / label descriptions for each dataset
>
> We do not prescribe a specific prompt format for datasets, to enable evaluation of a broad range of techniques, from those that do not use text formats to those that perform automatic prompt design. To ease model development, FLEET provides a lightweight conversion of task inputs to text format, but does not provide a prompt. Prompt-based model designers can take the raw text format of the examples returned from FLEET and convert them to their desired prompts, based on the task/dataset type. The provided Unifew baseline model is one example of how prompt-based models can work on FLEET.
>
> > concurrent work
>
> Thank you for the pointer. Zhong et al focus on zero-shot task transfer (one of our transfer types) and provide interesting analysis showing that meta-tuning and larger language models tend to help across a variety of held-out datasets and prompts. We will add this discussion to the related work section of the final version.

---

### Official Review · Reviewer_Z1sB · 2021-07-16

**Rating:** 7
**Confidence:** 3

**Summary:**

The authors of this paper present a few-shot NLP benchmark called FLEET. FLEET consists of a group of NLP datasets and evaluates four transfer settings: class transfer, domain transfer, task transfer, and pre-train transfer. The authors also provide a simple yet effective prompt-based learning baseline along with the FLEET benchmark.

**Limitations And Societal Impact:**

The authors provide a section addressing the limitations in the provided benchmark.

**Main Review:**

Strength:

1. A new few-shot benchmark with a variety of datasets and tasks. It provides separate evaluations for each transfer type and should be able to provide a comprehensive evaluation proxy for few-shot NLP frameworks.
2. Addresses several issues existing in previous few-shot NLP benchmarks, e.g. class imbalance and inconsistent problem settings.
3. A simple prompt-based learning framework with a decent performance on FLEET and other test sets that were used in previous work.

Weakness & Questions:

1. Currently, FLEET only contains one entity typing dataset (CoNLL). This dataset is only used for task & pre-train transfer. One could a) include more entity typing datasets b) and offer class transfer or domain transfer on the entity typing task.

**Time Spent Reviewing:**

5

---

> ### Author Response · Authors · 2021-08-10
> **Response (entity typing datasets)**
>
> Thanks for taking the time to review our paper and for your thoughtful comments.
>
> We selected the single entity typing dataset used by Bansal et al to enable comparison with them. We did not include additional entity typing datasets in meta-training, so that entity typing would remain a test of task transfer, as Bansal et al did. Leaving out a task as a test of task transfer necessarily precludes using that task to test domain or class transfer.
>
> We agree that there are many additional questions about few-shot transfer that should be asked in the future. We carefully designed our desiderata and framework tooling to provide a strong foundation for the creation of future benchmarks, and invite researchers to use our tools to specify new meta-training/testing splits, such as to test domain or class transfer for entity typing.

---

### Official Review · Reviewer_1wmg · 2021-07-16

**Rating:** 7
**Confidence:** 5

**Summary:**

The authors introduce FLEET a new benchmark for evaluating few-shot learning. It's focused on classification tasks in NLP. The features of this new benchmark are:
- evaluation of 4 different types of transfer
- textual labels for zero-shot classification
- experimentally determined sample size

In addition, they propose a new strong baseline called UniFew which is prompt-based model built on top of UnifiedQA. They show that this method is competitive with other few-shot systems. Finally, they release the code and framework for further extension.

**Ethical Concerns:**

None.

**Limitations And Societal Impact:**

Yes they have with their short statement. They should have included considerations that go into selecting the "few" examples that the models fine-tune on. These need to be free of any bias.

**Main Review:**

Overall, this work is going to be useful for the few-shot learning community.

Positives:
- Coverage across 20 different classification tasks
- Evaluation of different transfer types like class transfer, domain transfer, etc.
- Incorporating learnings from other fields like computer vision.
- Proposal of a strong baseline which is evaluated on the benchmark and compared against other SOTA techniques.

Negatives:
- The benchmark only includes classification tasks and in English only. More content on how this can be expanded to multilingual settings and other task types would be useful.
- More details on how the benchmark will look and how it's been designed would have been helpful. Screenshots of the baselines and the metrics reported would be good.


Questions for the authors:
- Can existing benchmarks like GLUE or SuperGLUE be converted to few-shot benchmarks by creating few-shot versions of them? Wouldn't that be better since we will know both few-shot and fully supervised performance on the same set of tasks? It's a bit suboptimal for everyone to create a different set of tasks as a benchmark.

Minor comments:
- Include references about multilingual benchmarks: XTREME, XGLUE, XTREME-R
- Number of shots and classes: the numbers chosen here are a bit arbitrary and more concrete experiments could have been better.
- The list of tasks is in the Appendix. This is important and should have been included in the main paper


**Time Spent Reviewing:**

3

---

> ### Author Response · Authors · 2021-08-10
> **Response**
>
> Thanks for taking the time to review our paper and for your insightful comments and feedback.
>
> > More content on how this can be expanded to multilingual settings and other task types
> > Include references about multilingual benchmarks: XTREME, XGLUE, XTREME-R
>
> Thank you for the suggestion. We will clarify that our framework can easily accommodate additional task types, including those for other languages and tasks (as long as they are included in the Huggingface datasets library, which already supports important multi-lingual datasets such as XTREME, XGLUE and XTREME-R). Testing multilingual transfer is an important future direction, which requires new work to determine appropriate meta-training/testing splits.
>
> > Details on how the benchmark will look
>
> We report full detailed benchmark results in the supplementary material (Appendix G), as computed by our benchmark code. We will clarify that the public leaderboard also provides a tabular view of these scores (one row per submission), and enables download of raw per-episode scores on request (e.g., for pairwise statistical testing). We will provide an illustrative screenshot in the appendix.
>
> > Can existing benchmarks like GLUE or SuperGLUE be converted to few-shot benchmarks by creating few-shot versions of them? Wouldn't that be better since we will know both few-shot and fully supervised performance on the same set of tasks?
>
> Our framework could be configured to use (Super)GLUE for meta-testing, but this wouldn’t enable testing all transfer types (e.g., none of the GLUE datasets test class transfer) and also wouldn’t enable comparison to prior meta-learning and few-shot evaluation suites. The datasets we selected were mainly introduced in a fully supervised setting, so it is possible to obtain upper bounds on SOTA performance.
>
> > They should have included considerations that go into selecting the "few" examples that the models fine-tune on. These need to be free of any bias.
>
> These examples are uniformly sampled in each episode, so they represent an unbiased estimate of performance on a random few-shot training set. We acknowledge that these samples may contain social and other biases associated with collection of the underlying datasets, and will elaborate in the final version.
>
> > Number of shots and classes: the numbers chosen here are a bit arbitrary and more concrete experiments could have been better.
>
> We followed Bao et al in limiting the number of classes and the maximum number of shots to 5 (some datasets have fewer than 5 classes due to having fewer than 5 classes total). We also followed other prior work in CV (​​Triantafillou et al) in sampling the number of shots uniformly at random. We believe these choices help to make FLEET more realistic and challenging than prior few-shot NLP evaluation suites. We mentioned these choices briefly in Section 4.3 but will clarify in the final version.
>
> > The list of tasks is in the Appendix
>
> Section 4.2 does list the meta-testing datasets, but we will more completely list all datasets in the main paper in the final version.

---

### Official Review · Reviewer_VPZK · 2021-07-19

**Rating:** 7
**Confidence:** 4

**Summary:**

Fleet is a new benchmark and leaderboard that combines previous work on few-shot text classification. While there are no new tasks, the new benhmark adds value by combining complimentary lines of work in a well designed evaluation framework.


**Limitations And Societal Impact:**

The authors point out that Fleet is limited by it's restriction to text-classification tasks. It is clearly not a defnitive benchmark, but still valuable.

The paper hints at the design of templates for the UniFew baseline but does not provide enough details for replication.


**Main Review:**

This paper presents a new benchmark and associated leaderboard that aggregates text classification tasks from three previous works on few-shot learning.

The tasks are chosen to cover a range of domain, task, and learning transfer. There is a detailed investigation of the trade-off between large test sets with few test episodes, and smaller test sets with many test episodes. The test setting is chosen to maximise precision of the benchmark given a moderate compute budget.

The paper presents a new prompt-based approach to few-shot learning that converts each of the classification tasks into a multi-choice QA task. The details of this conversion are not given in the paper and I would like to see them listed. Comparisons are also made to previous models that were applied to some subset of the tasks included in Fleet.

While this paper does not introduce any new tasks or evaluation methodologies, it does add value by combining several lines of complimentary work on few-shot learning into a single evaluation framework that has been well designed. The prompt based UniFew baseline also seems both simple and effective, although I would like to see more details of the prompt template design in the main paper.


**Time Spent Reviewing:**

2.5

---

> ### Author Response · Authors · 2021-08-10
> **Response**
>
> Thanks for taking the time to review our paper and for your thoughtful review. We are glad that you found our new benchmark valuable and the prompt-based model effective.
>
> > I would like to see more details of the prompt template design in the main paper...The paper hints at the design of templates for the UniFew baseline but does not provide enough details for replication.
>
> Further details about Unifew, including the exact prompt templates used for each task are already included in the appendices C and D (this is mentioned in lines 280-281 in the paper). We also have included source code to facilitate replication. We agree that such details are important and we will further elaborate on them in the main paper.
>
> > Comparisons are also made to previous models that were applied to some subset of the tasks included in Fleet.
>
> We briefly discuss the experimental setup in lines 287-295. To enable fair comparison and compare with the exact reported numbers by the prior work, instead of running their models on a setting they were not originally designed for, we test UniFew on the exact datasets and data splits released by the authors when available (H-SMLMT and LM-BFF). Some of the test datasets used by prior works are in the meta-training split of FLEET (because they were used for meta-training by other work we compare with) so excluded them for fairness. For Distributional Signatures, exact unprocessed datasets were not available, so we tried reproducing their numbers and evaluation setting. We sample (balanced) episodes using our framework after matching their test settings (number of shots and classes, class splits, etc.) and reproduce their reported results to within 1% absolute difference using their model code; we use these episodes for our experiments.

---

> > ### Comment · Reviewer_VPZK · 2021-08-27
> > **Response**
> >
> > Thank you for the clarifications.

---

### Decision · Program_Chairs · 2021-09-27

**Decision:**

Accept (Poster)

**Comment:**

This paper introduces a family of multi-task benchmarks for few-shot learning in NLP. It appears to be built on existing data with relatively lightweight new methodological contributions, but reviewers agreed that it represents a useful tool and a clear improvement over current standard practice.

I'd urge the authors to add some discussion of the licences that apply to the datasets used. This issue came up only toward the end of reviewer discussion, and so won't influence our decision about the paper, but it's not currently clear whether the required data for the benchmark is guaranteed to remain legally available or whether private firms can use it.